# Pick Your Channel: Ultra-Sparse Readouts for Recovering Functional Cell Types

## Abstract

Clustering neurons into distinct functional cell types is a prominent approach to understand how the brain integrates information about the external world. In recent years, digitial-twins of the visual system based on deep neural networks (DNNs) have become the de facto standard for predicting neuronal responses to arbitrary stimuli. Such DNNs are designed with a common core that learns a representation of the visual input that is shared across neurons, and a neuron-specific readout that linearly combines the core outputs to predict single neuron responses. Here, we propose a novel way to learn an ultra-sparse readout that, instead of linearly combining the shared core features, learns to pick a single channel for each neuron. For retinal ganglion cells, we find that, unlike the previous unconstrained models, this ultra-sparse readout triggers the neural predictive model to innately learn functional cell types with minimal loss in predictive performance. Furthermore, we show that state-of-the-art adaptive regularization models are unable to find such single channels, and that applying strong regularization to encourage sparse channels not only deteriorates performance but also results in response shrinkage. When applied to primary visual cortex neurons, our model exhibits a larger drop in performance compared to the unconstrained model, perhaps indicating a more continuous organization of neuronal function.

## 1 Introduction

Characterizing neurons into distinct functional cell types is key in discovering how the brain integrates and organizes information about the external world. In the retina, the first stage of the visual system, functional cell types are well defined for its output neurons, the retinal ganglion cells, each transmitting distinct information about the visual scene to the brain (Baden et al., 2016). Existence of such distinct functional cell types for the primary visual cortex (V1), however, is unclear with many studies supporting the hypothesis of a rather continuous functional organization. (Ustyuzhaninov et al., 2022; Weiler et al., 2023; Tong et al., 2023; Nellen et al., 2025).

Recently, data-driven deep neural networks (DNNs) have become the standard approach for modeling stimulus-driven neuronal responses, particularly in vision (Cadieu et al., 2014; Batty et al., 2017; Klindt et al., 2017; McIntosh et al., 2016; Cadena et al., 2019; Kindel et al., 2019; Walker et al., 2019a; Zhang et al., 2018; Ecker et al., 2018; Sinz et al., 2018; Burg et al., 2021; Cowley & Pillow, 2020). Such neural predictive models exhibit a common modular architecture of a *core* shared among neurons and a neuron-specific *readout* (Antolík et al., 2016; Klindt et al., 2017). The core learns shared representations of visual stimuli, while the readout linearly maps the core output features to neural responses. The core-readout architecture was later extended to temporal dynamics (Sinz et al., 2018; Höfling et al., 2024; Turishcheva et al., 2023) and a more efficient readout (Lurz et al., 2020) – called Gaussian readout.

Recently, Wang et al. (2025) trained a 13-mice CNN-based model and showed that readout weight vectors of these "digital twins" can be used to capture biological phenomena beyond their training data, such as cell morphology. Readout vectors have also been applied to cluster mouse V1 cells into functional groups (Ecker et al., 2018; Ustyuzhaninov et al., 2019; 2022; Nellen et al., 2025). To cluster neurons into different functional cell types based on readout vectors, each type would ideally map onto a single channel in the network to foster interpretability of the learned features. Such sparsity is typically enforced through L1 regularization on the readout weights. However, previous

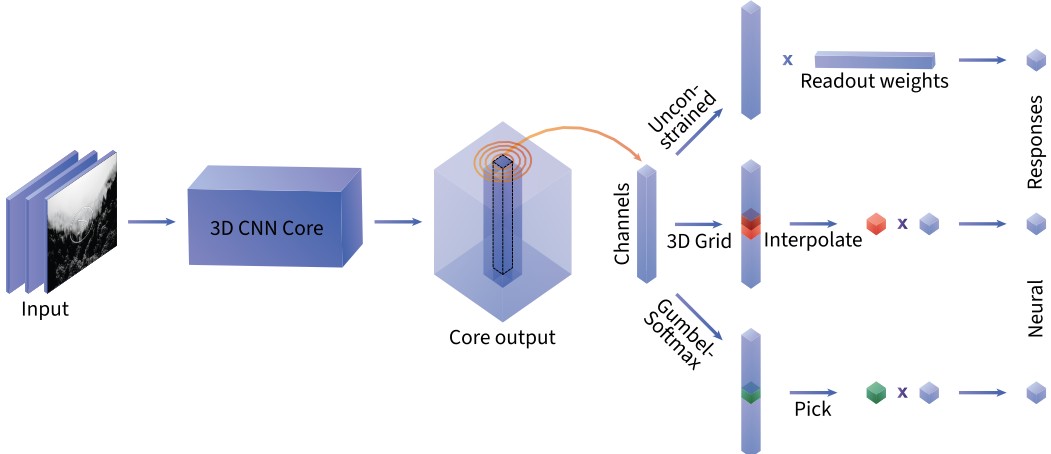

Figure 1: **Overview over our approach:** We evaluate different approaches for ultra-sparse readouts for core-readout neural encoding models. We use the same 3D CNN core architecture and a Gaussian readout to pick the spatial location of a neuron (orange circles). Once, a channel-dimensional vector is extracted at the neurons location in the visual field, we explore ultra-sparse readouts (mid and bottom right), and compare them against unconstrained readouts (top right).

work observed that too strong regularization causes shrinkage in the predicted responses towards the population mean (Turishcheva et al., 2024a), which is undesirable.

To address this problem, we propose a novel approach of an ultra-sparse readout that instead of linearly combining the core output features learns to pick a single channel per neuron to read out from. To this end, we explored three strategies to pick a single channel of the CNN per neuron:

- A **Gumbel-Softmax** readout where single channels are sampled from neuron-specific Gumbel-Softmax distributions (Jang et al., 2016; Maddison et al., 2016).

- A **3D Grid** readout, which is an extension of the Gaussian readout (Lurz et al., 2020) to the channel dimension. This readout is not strictly limited to a single channel, but may also interpolate at most two neighboring channels.

- A **REINFORCE** readout based on a policy gradient method where single channels are picked using neuron-specific channel selection policies (Williams, 1992).

We tested our ultra-sparse readout on recordings from mouse retinal ganglion cells and found that it innately identifies established functional cell types, while incurring only minimal loss in predictive performance. We also show that L1 regularization is unable to identify single channel readout vectors, while suffering both from performance loss and response shrinkage. We also tested our sparse models on mouse V1 neurons, where they fell short of the performance of unconstrained Gaussian readout models. This is expected and is consistent with the idea of a more continuous functional organization in V1 shown in prior work (Ustyuzhaninov et al., 2022; Nellen et al., 2025).

In summary, our ultra-sparse readouts introduce a plug-and-play modification of a Gaussian readout that encourages the model to innately group neurons into functional types and contributes to more interpretable models of the early visual system.

## 2 MODELS AND METHODS

### 2.1 NEURAL ENCODING MODELS

Our neural encoding models are based on the common CNN core-readout architecture design. Although recent developments extended neural encoding models from CNN to transformer or CNN-transformer hybrid architectures (Li et al., 2023; Lin et al., 2024; Saha et al., 2024; Pierzchlewicz et al., 2023), we focus on CNN architectures because they naturally handle spatiotemporal data from

natural movies, are well established for modeling early visual areas (retina and V1), and currently offer comparable performance to transformers, which remain less developed for video.

Our **core** is based on a previous architecture (Höfling et al., 2024; Turishcheva et al., 2023) and learns shared representations from visual and behavioral inputs. Behavioral parameters (locomotion speed and pupil dilation size) were included as additional uniform input channels to the core to capture modulation of neural population responses correlating with behavior (Reimer et al., 2014; Sinz et al., 2018; Niell & Stryker, 2010; Schröder et al., 2020; Stringer et al., 2019).

The core is a space-time separable 3D factorized Convolutional Neural Network. Within each layer, spatial and temporal convolutions are applied separately with kernel sizes treated as hyperparameters. The input convolutional kernels are regularized using Laplace regularization controlled by separate spatial and temporal factors. Each convolutional layer is followed by a batch normalization layer, tuned with a momentum hyperparameter, and a nonlinearity from a selection of ELU, Softplus, and ReLU. The exact choices for these hyperparameters for different instantiations of the models are specified in Appendix (Table 1 and 3). Apart from different choices of hyperparameters, the principal core architecture stays the same across different models. The core outputs a tensor $\mathbf{x} \in \mathbb{R}^{w \times h \times c}$ per time point, which represents the learned feature space.

The **readout** is neuron-specific, and maps the output of the core onto neuronal responses. Our base for the unconstrained and sparse models is the Gaussian readout introduced by Lurz et al. (2020). Following their method, the location of each neuron's receptive field is sampled from a 2D Gaussian distribution, parameterized with mean $\boldsymbol{\mu} \in \mathbb{R}^2$ and covariance $\boldsymbol{\Sigma} \in \mathbb{R}^{2 \times 2}$. Inspired by the retinotopic organization of visual brain areas, the means of the Gaussian distributions are initialized using a remapping of the anatomical coordinates of neurons recorded during experiments as introduced by Bashiri et al. (2021). During training, receptive field locations are sampled from neuron-specific distributions, while during evaluation they are fixed to the means. The receptive fields $(x, y)$ can also be shifted in accordance with the gaze/pupil position changes, using a separate shifter network (Sinz et al., 2018). The receptive field locations $(x, y)$ are then used to extract the core output features at a single spatial position via bilinear interpolation (Jaderberg et al., 2015; Lurz et al., 2020). This yields a channel dimensional vector $\mathbf{v} \in \mathbb{R}^c$ per time point.

All readout architectures that we explain below, use the same model until this point. They do, however, differ in the way the single dimensions from the extracted feature vector $\mathbf{v}$ are combined into a prediction of the neuronal response (Figure 1).

**Unconstrained model.** The classical unconstrained models linearly combine the extracted features using neuron-specific learnable readout weights $\mathbf{w}^\top \mathbf{v}$. The readout weights $\mathbf{w}$ are $L_1$ regularized to encourage sparsity. Unless we use an adaptive readout (Turishcheva et al., 2024a), we use $\gamma = 1$ applied uniformly for all neurons.

**Adaptive Regularization model.** Turishcheva et al. (2024a) introduce an adaptive regularization for the readout vector $\mathbf{w}$, for which each neuron's regularization strength is a learnable parameter. Global regularization strength is controlled by $\gamma$. The individual coefficients are controlled by a log-normal prior, for which a hyperparameter $\sigma$ controls how far they deviate from the overall mean.

**Ultra-sparse readout: Gumbel-Softmax.** Our Gumbel-Softmax model implements an ultra-sparse readout that picks a single channel from the extracted features, instead of linearly combining the channels. Single channels are sampled from neuron-specific Gumbel-Softmax distributions, introduced in Jang et al. (2016); Maddison et al. (2016). This continuous distribution allows an approximation of discrete categorical samples, and thus a reparametrization trick. We set the number of categories equal to core output channels $c$. During the forward pass, the categorical samples are $c$-dimensional one-hot vectors, effectively picking a single channel.

We learn one Gumbel-Softmax distribution for each neuron and tune it with a common temperature parameter $\tau$. When $\tau$ is high, the distributions become smoother, approximating uniform values. When $\tau$ is low, the distributions become sharper, approximating categorical distributions. As we would like to encourage exploration of different channels at the beginning of training and to gradually converge to a single channel choice towards the end, we implement a cosine scheduler for $\tau$, that goes from $\tau = 10$ to $\tau = 0.5$ over $T$ epochs.

**Ultra-sparse readout: 3D Grid.** Our 3D Grid model is an extension of the Gaussian readout idea to channels. This sparse readout does not necessarily pick a single channel from the extracted core

output features, but linearly interpolates between at most two neighboring channels in the channel dimension. This is similar to how 2D Gaussian readout interpolated in $(x, y)$ dimensions to extract the core output features at a spatial position (Lurz et al., 2020). We introduced a learnable parameter $z$ that models the "location" in the channel dimension. As the interpolation is handled using grid coordinates of range $[-1, 1]$, we initialized $z$ uniformly in a small range $[-0.1, 0.1]$, and additionally constrained with a $tanh$ nonlinearity.

**Ultra-sparse readout: REINFORCE.** Our REINFORCE readout implements an ultra-sparse readout that picks a single channel similar to the Gumbel-Softmax model. However, in this version of sparse readout we use a policy gradient method: the REINFORCE algorithm (Williams, 1992). For each neuron $n$ we learn a discrete policy $\pi_n$ over channels via softmax of a learned parameter vector per neuron. The logits of each policy are initialized randomly from a $\mathcal{N}(0, 0.01)$. During training we sample channels from the policy probabilities using a multinomial distribution. We use the log probabilites of selected channels, $c_n$ for computing the REINFORCE loss term with a "detach" trick to get the correct gradient with auto-differentiation. In addition to the per-neuron Poisson loss, $\ell_n$, we use a neuron-specfic moving-average baseline, $b_n$, for variance reduction. This yields the final loss term $\mathcal{L}_{\text{reinforce}} = \sum_n \left[\ell_n - b_n\right]_{sg} \log \pi_n(c_n)$ where $\left[\ell_n - b_n\right]_{sg}$ is the advantage term inside a stop gradient operation to treat it as a constant during optimization. Similar to the Gumbel-Softmax model, we encourage exploration in the early stages of training (up to 40 epochs), with an additional entropy regularizer that is scaled dynamically to match the scale of the REINFORCE loss. If $H(\pi_n) = -\sum_c \pi_n(c) \log \pi_n(c)$ is the entropy of neuron $n$'s policy, and $\widehat{R}$ is an exponential moving average of the absolute REINFORCE loss magnitude, then we compute $\beta_{\text{entropy}}^{\text{dyn}} = \frac{\widehat{R}}{\left|\sum_n H(\pi_n)\right| + \varepsilon}$, at each iteration, where $\varepsilon$ is a small constant for numerical stability. $\beta_{\text{entropy}}^{\text{dyn}}$ is treated as a constant, i.e. does not propagate gradients. The entropy term in the total objective is then given by $\mathcal{L}_{\text{entropy}} = \beta_{\text{entropy}}^{\text{dyn}} \left(-\sum_n H(\pi_n)\right)$. We use a separate optimizer with a fixed learning rate to learn the policies for channel selection.

In all ultra-sparse models, a neuron-specific learnable scale and bias term are applied at the end of the readout. The results are put through an ELU nonlinearity and offset by 1 to ensure positive output neural responses.

## 2.2 MODEL TRAINING

Our models are trained to minimize a Poisson loss, with early stopping and learning rate schedulers similar to previous neural encoding models (Lurz et al., 2020; Höfling et al., 2024; Turishcheva et al., 2023). Training hyperparameters for the models with retinal data are given in Appendix Table 2, and in Table 4 for primary visual cortex models.

## 2.3 DATA

**Retinal ganglion cell axons.** We used a large-scale two-photon imaging dataset from *in vivo* mouse retinal ganglion cell axon endings measured in the superior colliculus of awake, head-fixed mice. This dataset is similar in structure to the public primary visual cortex dataset below (Turishcheva et al., 2023). Notably, it has a set of unique natural movies used for training our models, and 6 natural movies that were shown repeatedly, used to test the predictive performance of our models. Each natural movie is $10s$ in length. We also used cell responses to simple synthetic light stimuli (chirp and moving bars) commonly used to identify functional cell types (Baden et al., 2016). The chirp stimulus contains a bright full-field white step stimulus with increasing frequency and contrast components that were modulated by two sinusoids. The moving bar stimulus is a bright bar moving in eight directions. Both synthetic stimuli are $32s$ long, and have repeating trials. This dataset contains quality-controlled 3,175 axonal boutons, which we will refer to as neurons.

**Primary visual cortex.** For the primary visual cortex experiments, we used a public dataset *29156-11-10* from the dynamic Sensorium 2023 competition. Detailed description of data is provided by the white paper of Turishcheva et al. (2023). Importantly, we measure performance on the *live main test set*.

## 2.4 Metrics of model performance

Our neural encoding models predict neuronal responses given videos and behavior as input. We use similar measures of predictive performance, as Turishcheva et al. (2023). To assess the model's performance in capturing stimulus-specific components of neural responses, we use **Correlation to Average**. Correlation is computed per neuron $n$ across stimuli and time, between responses and predictions averaged across repeated presentations of the same stimulus, $\bar{r}_n$ and $\bar{p}_n$, respectively, $\rho_{\text{ta}} = \text{corr}(\bar{r}_n, \bar{p}_n)$. To assess the model's performance in capturing trial-to-trial variability in neural responses, we use **Single Trial Correlation**. Here, correlation is computed between individual trial responses $r_{nk}$ and predictions $p_{nk}$ across time and stimuli: $\rho_{\text{st}} = \text{corr}(r_{nk}, p_{nk})$.

## 2.5 Analysis

**Consistency of channels**  To measure the consistency of responses within a single readout channel, we compute the correlation of single neuron's activity to the mean activity of all neurons that pick this channel. For this we concatenate responses to chirp and moving bar. To not confound the correlation by including the single trial in the mean computation, we use a leave-one-out jackknife estimator: $\text{Consistency}(n) = \text{corr}\left( r_n, \ (|S_n| - 1)^{-1} \sum_{\substack{m \in S_n \\ m \neq n}} r_m \right)$ where $r_n$ is the response of neuron $n$ to a particular stimulus, $S_n$ is the set of neurons that pick the same channel as neuron $n$, and $r_m$ are the responses of these neurons to the same stimulus. This measures the consistency of the neuronal responses within a channel.

In addition, we used Adjusted Rand Index (ARI) to measure how consistently our ultra-sparse readout model identifies the same channels for two neurons across model initializations (Pedregosa et al., 2011; Hubert & Arabie, 1985). ARI scores are adjusted for chance and measure similarity between cluster labels. Cluster labels in our case are selected channels in the readout.

**Sparseness of readout weights**  We measured the sparseness of unconstrained readouts with entropy of readout weights. For this we first normalized the absolute readout weights $\mathbf{w}$ to get a probability distribution per neuron $n$: $\mathbf{p}_n = |\mathbf{w}_n|/\|\mathbf{w}_n\|_1$ We then calculated the entropy per readout weight $H[\mathbf{p}_n] = -\sum_i p_{ni} \log p_{ni}$. If the readout focuses on a single channel, the entropy is $H[\mathbf{p}_n] = 0$. If the weights spread uniformly across $c$ channels, it is maximal with value $H[\mathbf{p}_n] = \log c$.

**Response shrinkage**  To assess the effect of high regularization on response shrinkage in the Adaptive Regularization model, we obtained the predictions of models trained with different regularization strengths $\gamma$ to the chirp stimulus (see section 2.3). We first averaged the predictions across repeated presentations of the same stimulus, and then computed the variance of these mean predictions around the mean across time per neuron, and finally report the mean across neurons. If the predicted response shrinks with higher regularization, this variance decreases.

**Identifying known functional cell types from chirp and moving bar responses**  To match readout channel responses with identified cell types, we used maximum Spearman correlation. We correlated the cell type responses to chirp stimulus and moving bars from Baden et al. (2016) with the mean channel responses from our sparse readout model. The best matching cluster was identified by calculating the mean correlation across both chirp and moving bar responses.

## 3 Experiments and Results

**Gumbel-Softmax readout almost matches performance of unconstrained model on retinal data**
We compared the predictive performance of unconstrained neural predictive models with sparse readout models trained on retinal ganglion cell dataset. The classical model with an unconstrained readout from Lurz et al. (2020), and its more recent improvement with adaptive regularization from Turishcheva et al. (2024a) reached a similar predictive performance and performed the best among all models due to representational flexibility, as expected. When we constrain the readout to be ultra-sparse, our model with Gumbel-Softmax sampling outperformed other implementations based on 3D grid interpolation and REINFORCE algorithm (Figure 2A). The Gumbel-Softmax model closely

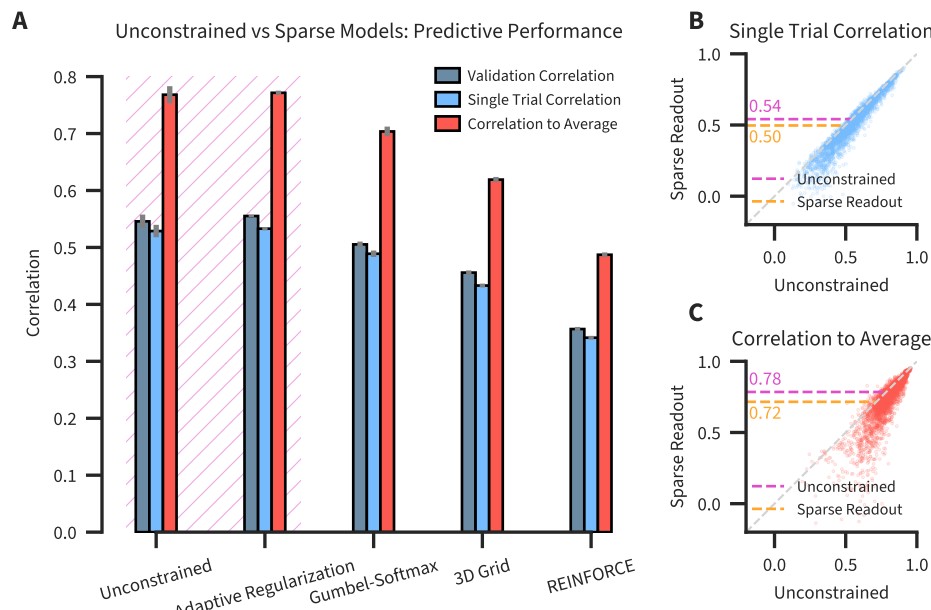

Figure 2: Gumbel-Softmax Readout almost matches the performance of unconstrained models on retinal data. **A** Correlations of unconstrained and sparse readout models on validation and test sets averaged over neurons. Validation correlation is also a single trial correlation, but over the input stimuli in the validation set, while the latter is for repeated trials of input stimuli present in the test set. Correlation to average was computed on the test set. Error bars indicate standard deviations of correlations across 5 seeds. Shaded region indicates unconstrained readouts. **B-C** Scatter plots of single trial correlations and correlations to average per neuron of the best across seeds unconstrained vs the Gumbel-Softmax model.

matched ($\approx 90\%$) the performance of the classical unconstrained model in predicting single trial responses and trial-averaged responses, thereby capturing both trial-to-trial variability and visual stimulus specific components of neural responses (Figure 2B,C), despite being constrained to a single channel only.

$L_1$ **regularized models fail to identify single channel**  Next we tested whether the unconstrained models could in principle learn to pick single channels through regularization of readout weights (Figure 1). To that end, we used the model with adaptive regularization and the core module of our Gumbel-Softmax ultra-sparse model, since we know that a single channel readout can achieve good performance with that core. We then first selected the $\sigma$ hyperparameter of the adaptive regularization readout, such that the per-neuron regularization coefficients are distributed broadly with mean closer to 1 (Figure 3A). The spread of the distribution ensure that regularization is indeed adaptive, following the results and guidelines from Turishcheva et al. (2024a). Based on this, we fixed $\sigma = 0.16$ and trained neural predictive models with varying global regularization strength $\gamma$. We then extracted the readout weights of the trained models and measured their sparseness with entropy. Adaptive regularization models were not able to find sparse readout weights even with high regularization as the readout weight entropy never reached zero (Figure 3C).

Somewhat unexpectedly, as we increased the strength of regularization, the entropy of readout weights decreased at first, and then started increasing. This counterintuitive effect of regularization on sparseness can be explained by observing that the norm of readout weights decrease with increasing $\gamma$, as expected (Figure 3E). In the limit, the weights are very close to zero and become noisy. When normalizing them for the entropy computation, this is reflected in the increased entropy.

Moreover, at high regularization strengths the unconstrained model substantially suffered in performance (Figure 3B), and the model output predictions shrunk towards the mean of predictions (Figure 3D). Overall, this demonstrates that unconstrained models are not able to find sparse chan-

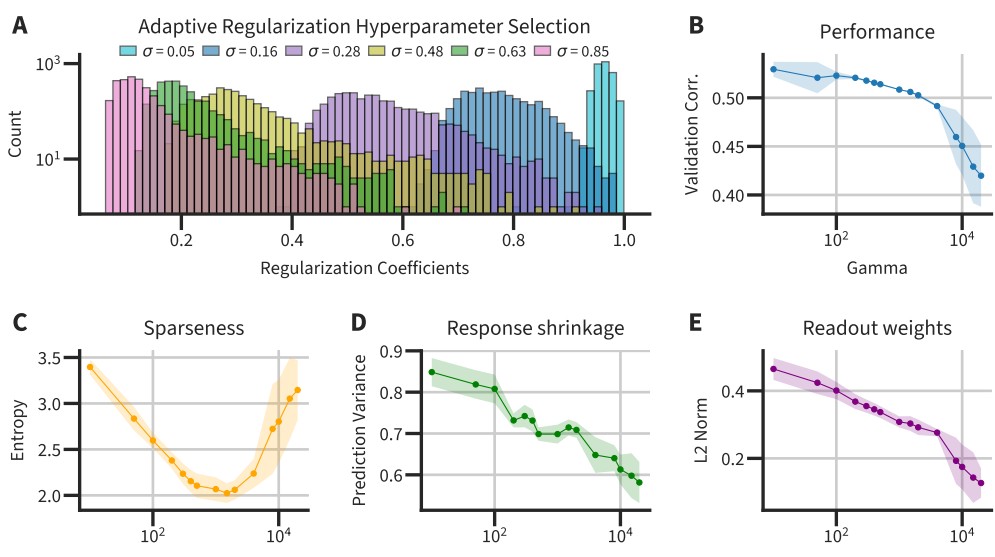

Figure 3: Adaptive Readout cannot find sparse readout channels at all, and decreases performance with high regularization and leads to response shrinkage. **A** Hyperparameter selection for the adaptive regularization readout. The plot shows the distributions of per-neuron regularization coefficients at different levels of $\sigma$. The global regularization strengths $\gamma$ were chosen randomly during the hyperparameter search, and are in the range from $[10 - 20]$. **B-E** Analysis on neural predictive models trained at different levels of regularization strength $\gamma$. All models were trained with 5 seeds and the shaded regions indicate $\pm 1\, std$. **B** Validation correlation of models. **C** Entropy of readout weights. **D** Variance of mean predictions (averaged across trials) around the mean over time. **E** $L_2$ norm of readout weights.

nels with strong regularization even when given the best options to achieve that (the ultra-sparse core). Higher regularization mainly results in loss of performance and response shrinkage.

**Ultra-sparse readout consistently identifies cell types** After we identified the best sparse readout based on Gumbel-Softmax sampling, we tested whether the channels that neurons learned to pick are consistent in terms of their responses (Figure 1). For this we took the mean responses of all neurons of a channel to chirp and moving bar stimuli, commonly used to identify functional cell types for retinal ganglion cells (Baden et al., 2016), and measured their consistency with a jack-knifed correlation of the single neuron responses against the group mean (see Analysis). We found that the responses of neurons within the sparse readout channels are highly consistent (Figure 4A). Furthermore, the channel labels for neurons learned by our sparse readout across different model initializations were also highly consistent, as measured by Adjusted Rand Index scores (Figure 4B). While the ARI scores of our sparse readouts might be lower than those of dedicated clustering algorithms, we report high functional consistency within readout channels, important for recovering functional cell types. Furthermore, the ARI scores reported for our sparse readouts might be underestimated, due to the fact different readout channels might represent the same functional cell type. We have done an approximate measure of how ARI would increase, if our readout had smaller number of channels (Appendix Figure S1). Finally, a high ARI score only implies that repeated runs of the clustering yield similar results. It does not measure the goodness or even separation of the clusters.

Results of our sparse readout shows that the model learned to group neurons into channels based on their functional responses (Figure 4A). Next, we tested whether the mean responses of these channels correspond to known functional cell types in retinal ganglion cells, as previously identified in Baden et al. (2016) using parametric chirp and moving bar stimuli. Using our sparse model, we were able to recover a diverse set of functional response types, including direction selective, non-direction selective, on transient, on sustained, off transient, off sustained, on–off, and contrast-

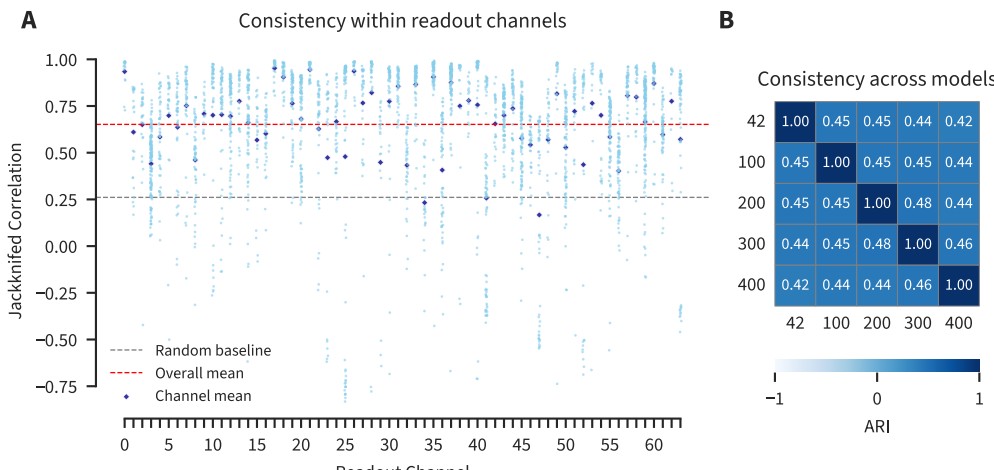

Figure 4: Ultra-sparse model picks functionally consistent channels across model initializations. **A** Consistency of responses within readout channels measured with jackknifed correlation. The results are shown for the best performing Gumbel-Softmax model seed. The overall mean (red dashed line) is the consistency correlation averaged across channels and model seeds. The random baseline (grey dashed line) is a consistency correlation (averaged across seeds) of a random group with number of elements equal to average readout channel size. **B** Consistency of picked readout channel labels across different seeds measured with Adjusted Rand Index.

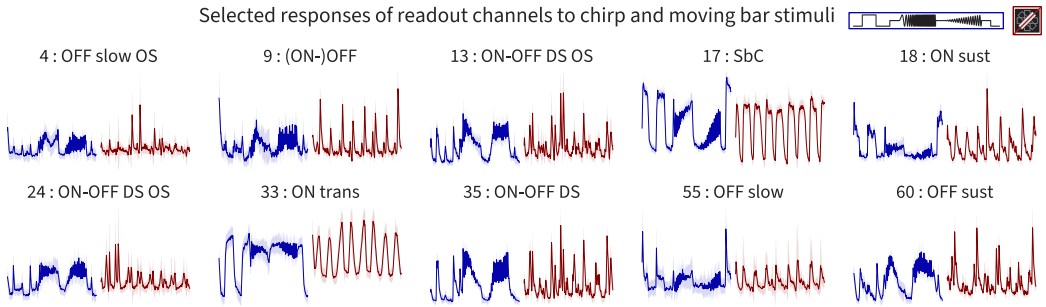

Figure 5: Responses of readout channels correspond to known functional cell types in retinal ganglion cells such as direction selective (DS) cells, non-direction selective cells, ON transient (trans), ON sustained (sust), OFF trans., OFF sust., ON-OFF responses, and neurons that are suppressed by contrast (SbC). These functional cell types are identified using chirp and moving bar stimuli (schematics on the top right). Solid lines represent means of all neurons grouped into the readout channel. And shaded regions are ± standard deviations.

suppressed responses. These categories map well onto distinct retinal ganglion cell types identified in earlier work (Figure 5).

**Application to Primary Visual Cortex**   So far we have validated that our sparse readout models can learn to identify functional cell types with minimal loss in performance over unconstrained models using data from retinal ganglion cells. While we know that there are clearly defined cell types in the retina, the existence of functional cell types in the primary visual cortex (V1) is not as clear. Prior work suggests that there are no distinct cell types in V1 and neurons rather form a continuum of functions(Ustyuzhaninov et al., 2022; Weiler et al., 2023; Tong et al., 2023; Nellen et al., 2025). Hence, we wanted to test whether our sparse readout model can rival the performance of state-of-the-art unconstrained models for V1, which would indicate a cluster structure. We trained our sparse readout models on data from V1, and found that the ultra-sparse model based on Gumbel-Softmax sampling outperforms the interpolation based 3D Grid model (Figure 6). However, when

comparing the best sparse model against the unconstrained readout model, we found that the sparse readout suffers a larger performance hit ($\approx 40\%$ loss) in V1 models compared to retinal models. This could be evidence that functional cell types are not as clearly separated in V1 as they are in the retina, and that a continuum of functional cell types in V1 could be an alternative explanation.

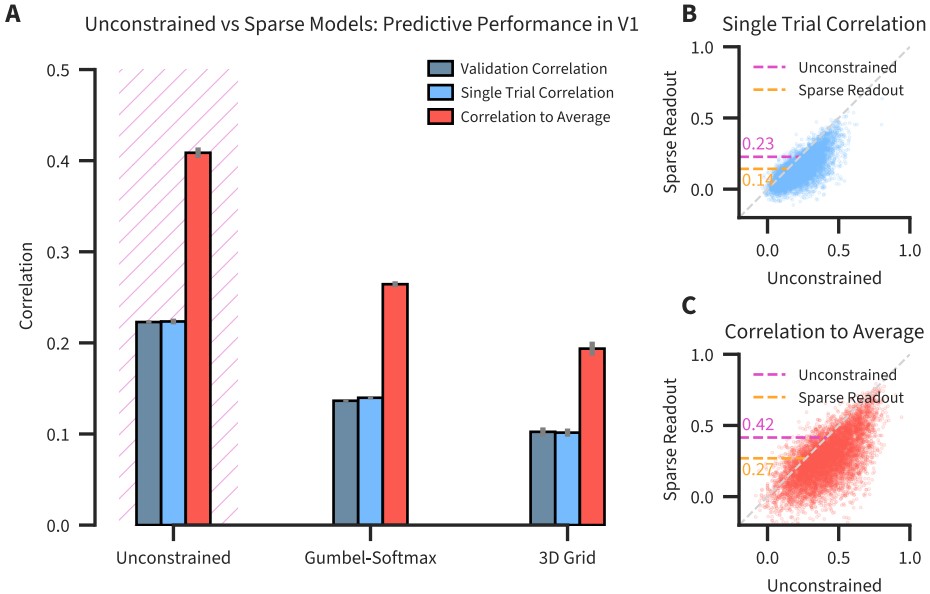

Figure 6: Application of sparse readouts to V1 neurons. **A-C** Legends identical to Figure 2

## 4 SUMMARY AND DISCUSSION

We explored different novel ultra-sparse readouts that trigger neural predictive models to innately learn functional cell types with minimal performance loss. We found that the Gumbel-Softmax implementation performed best across datasets. While the method successfully identified consistent retinal ganglion cell types, the larger performance drop on V1 neurons suggests it may be best applicable to brain areas with discrete rather than continuous functional organization – by design. In that case our sparse readouts can offer improved biological interpretability by mapping readout channels to functional cell types in neural encoding models.

There are also several limitations to our current approach. First, the ultra-sparse readout is currently restricted to selecting exactly one channel per neuron, which may be overly restrictive for neural systems that genuinely integrate information across multiple feature dimensions. Secondly, as the true number of functional cell types is oftentimes unknown, if we set the number of core output channels for our models high, then the same functional cell type might be represented at multiple readout channels. Thus, ideally one would need to empirically detect a minimum number of readout channels that gives a desirable tradeoff between predictive performance and consistent readout channels. Third, the consistency of channel selection across model initializations, while good, is not perfect, although this does not reflect the quality of the clusters which seem to be good based on visual inspection and our consistency measure.

Future work could extend this framework in several promising directions. The ultra-sparse constraint could be relaxed to allow selection of a fixed small number of channels (e.g., 2-5), potentially revealing how multiple features combine hierarchically in visual processing. The identified feature channels could serve as a foundation for generating synthetic stimuli that maximally excite specific cell types, providing a powerful tool for experimental neuroscience (Walker et al., 2019b; Bashivan et al., 2019). Furthermore, the approach could be applied to study developmental changes in functional organization or to investigate how cell type specialization emerges across different species.

Finally, incorporating temporal dynamics into channel selection could reveal how functional cell types adapt their feature preferences based on behavioral context or stimulus history.

## REPRODUCIBILITY STATEMENT

Code implementation required to reproduce the experiments presented in this paper, including an Apptainer container for easy setup, will be made available in a public repository upon acceptance. All relevant model and training hyperparameters are provided in the appendix for full transparency. The experiments were conducted using Python 3.9, PyTorch Version: 1.13.1+cu117, CUDA Version: 11.7. All models were trained with A100 GPUs.

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

## APPENDIX

### HYPERPARAMETERS FOR MODELS AND TRAINING

We conducted a hyperparameter search for each model type separately to give a fair chance to find the best performing models. The selection of hyperparameters to tune were identical across models, except for the training hyperparameters that were specific to the model. This includes the max epochs for cosine scheduler of temperature $\tau$ in the Gumbel Softmax model, and baseline momentum and learning rate of channel policies in the REINFORCE model. In the readout of the V1 models, we additionally tuned the hyperparameters for Gaussian sampling of spatial positions: initial $\mu$ and $\Sigma$ of each model. As we are modeling a different visual area with potentially different receptive fields of neurons, tuning these hyperparameters gives a fair chance for V1 models to improve predictive performance.

Table 1: Hyperparameters for the factorized 3D convolutional cores of retinal ganglion cell models

| Hyperparameter | Unconstrained | Adaptive Reg | Gumbel-Softmax | 3D Grid | REINFORCE |
|---|---|---|---|---|---|
| Number of Layers | 3 | 3 | 4 | 4 | 4 |
| Hidden Channels (per layer) | 64, 64, 64 | 128, 64, 64 | 32, 32, 64, 64 | 16, 32, 64, 64 | 32, 32, 64, 64 |
| Spatial Input Kernel Size | 11×11 | 3×3 | 7×7 | 4×4 | 17×17 |
| Temporal Input Kernel Size | 11 | 11 | 3 | 20 | 7 |
| Spatial Hidden Kernel Size | 5×5 | 5×5 | 3×3 | 6×6 | 4×4 |
| Temporal Hidden Kernel Size | 5 | 14 | 7 | 7 | 10 |
| Activation Function | ELU | ELU | Softplus | ELU | Softplus |
| Spatial Regularization ($\gamma_{spatial}$) | 10.0000 | 21.0403 | 0.2456 | 18.7300 | 21.6397 |
| Temporal Regularization ($\gamma_{temporal}$) | 0.0100 | 0.4043 | 0.0149 | 0.1652 | 0.4733 |
| Batch Normalization Momentum | 0.7000 | 0.4575 | 0.7442 | 0.7906 | 0.6494 |

Table 2: Training hyperparameters for the Retinal Ganglion Cell models.

| Hyperparameter | Unconstrained | Adaptive Reg | Gumbel-Softmax | 3D Grid | REINFORCE |
|---|---|---|---|---|---|
| LR decay steps | 8 | 6 | 8 | 8 | 6 |
| LR decay factor | 0.3 | 0.5264 | 0.3 | 0.3202 | 0.5908 |
| LR initial | 0.0050 | 0.0043 | 0.0068 | 0.0174 | 0.0156 |
| Optimizer | AdamW | AdamW | Adam | AdamW | Adam |
| Max epochs for $\tau$ scheduler | - | - | 75 | - | - |
| Baseline momentum | - | - | - | - | 0.8621 |
| LR channel policies | - | - | - | - | 0.3232 |

### CONSISTENCY MEASURES

We performed hierarchical clustering of mean readout channel responses from Figure 4. We first computed correlation distance and a linkage matrix between the mean readout channel responses. Then we assigned cluster labels for each readout channel (*scipy* pdist, linkage, fcluster with specified number of clusters) (Virtanen et al., 2020). This allowed us to obtain hierarchical clusters from readout channels with number of clusters in the range from 1 to 64. Then we mapped the neurons from the readout channels to the hierarchical clusters, and computed the consistency of these clusters using Adjusted Rand Index and Jackknife correlation (Figure S1).

Table 3: Hyperparameters for the core and readout of V1 models

| Hyperparameter | Unconstrained | Gumbel-Softmax | 3D Grid |
|---|---|---|---|
| Number of Layers | 3 | 3 | 5 |
| Hidden Channels (per layer) | 128, 128, 128 | 128, 128, 128 | 32, 64, 128, 64, 32 |
| Spatial Input Kernel Size | $14\times14$ | $12\times12$ | $10\times10$ |
| Temporal Input Kernel Size | 7 | 14 | 12 |
| Spatial Hidden Kernel Size | $8\times8$ | $9\times9$ | $5\times5$ |
| Temporal Hidden Kernel Size | 9 | 5 | 10 |
| Activation Function | ReLU | ELU | ELU |
| Spatial Regularization ($\gamma_{\text{spatial}}$) | 19.2068 | 8.9025 | 25.7434 |
| Temporal Regularization ($\gamma_{\text{temporal}}$) | 0.1919 | 0.2305 | 0.3625 |
| Batch Normalization Momentum | 0.3153 | 0.2885 | 0.4621 |
| Readout: Initial $\mu$ | 0.9087 | 0.0810 | 0.2211 |
| Readout: Initial $\Sigma$ | 0.6664 | 0.6520 | 0.6928 |

Table 4: Trainer hyperparameters for the V1 models.

| Hyperparameter | Unconstrained | Gumbel-Softmax | 3D Grid |
|---|---|---|---|
| LR decay steps | 3 | 8 | 7 |
| LR decay factor | 0.5716 | 0.3416 | 0.5925 |
| LR initial | 0.0021 | 0.0137 | 0.0074 |
| Optimizer | AdamW | Adam | AdamW |
| Max epochs for $\tau$ scheduler | - | 64 | - |

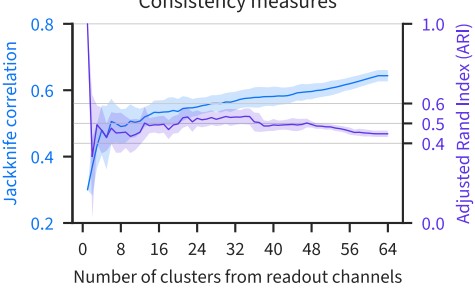

Figure S1: Consistency of hierarchical clusters from readout channels. Adjusted Rand Index for consistency of labels. Jackknife correlation for consistency of responses within clusters of readout channels. Shaded regions indicate standard deviation across 5 seeds.

CLUSTERING READOUT WEIGHTS OF UNCONSTRAINED MODELS

We compared the clustering consistency of our sparse readout to dedicated clustering methods. We performed Gaussian Mixture Model (GMM) and K-means clustering on the readout weights of the better performing unconstrained model. We set the number of clusters in both cases to be 64, same as the number of readout channels in the sparse model. Using the resulting labels from GMM and K-means clustering, we computed the consistency of these clusters using Adjusted Rand Index and Jackknife correlation (same as Figure 4). We report that clustering the readout weights of unconstrained model resulted in higher cluster consistency than from the readout channels of our sparse model (Table 5).

Table 5: Clustering metrics for the GMM and K-means on readout weights of unconstrained models. Values shown are means over 5 models with different seeds.

| Metric | Gumbel-Softmax | Unconstrained GMM | Unconstrained K-means |
|---|---|---|---|
| ARI | 0.448 | 0.599 | 0.613 |
| Jackknifed correlation | 0.652 | 0.757 | 0.767 |

This result intrigued us whether we can predict neural responses posthoc from the unconstrained model using the cluster mean readout weights. We first grouped the neurons in the unconstrained model based on the labels obtained from GMM and K-means clustering the readout weights. Then for each group we computed the mean readout weight, and replaced the neuron-specific readout weights with the cluster mean. To find the right scale for each neuron we divided the mean readout weight with its norm, and multiplied with the norm of the neuron-specific "old" readout weight. From this we computed the predictive performance of the unconstrained models, where neuron-specific readout weights are replaced with their cluster mean readout weights (Figure S2). We report that the predictive performance of these models are practically the same as our sparse models, and show similar minimal loss in predictive performance over the unconstrained model. This results in yet another method to achieve our goal of finding functional cell types in neural encoding models. Unlike the sparse model, this new method is posthoc and required manually changing the model parameters.

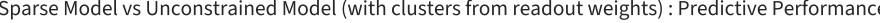

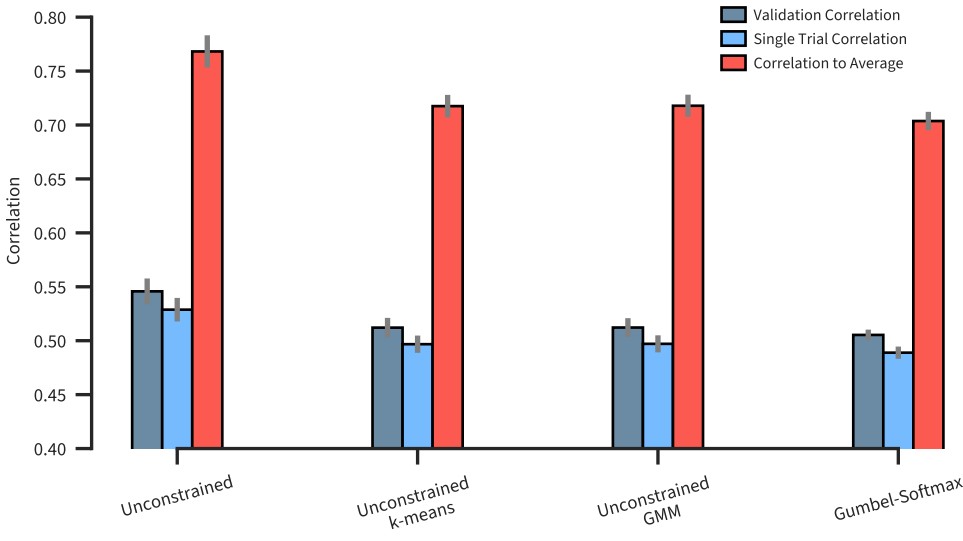

Figure S2: Predictive performance of unconstrained models with clusters from readout weights. Legends identical to Figure 2A

ULTRA-SPARSE MODEL WITH A TRANSFORMER CORE

To show that our ultra-sparse readout remains applicable to other architectures besides CNNs, we exchanged the core of our models to a Video Vision Transformer (Arnab et al., 2021). Specifically, we adapted the transformer core of the ViV1T model, which was among the winners of the neural encoding dynamic Sensorium challenge (Turishcheva et al., 2024b; Li et al., 2023). We replaced our previous CNN core with a video transformer, where the core embedding dimension becomes the equivalent of CNN feature channels. We tested the performance of unconstrained and ultra-sparse (Gumbel-Softmax) readout models (Table 6). The reported performances are from the best performing models, found through a search for suitable architecture and training hyperparameters (Table 7). We report that the unconstrained and sparse (Gumbel-Softmax) models with a transformer core achieved performance slightly lower to the models with CNN cores. Similar to before, the unconstrained models, given the same architecture of the core, outperform the sparse models (Table 6). Crucially, however, the ultra-sparse model with the transformer core consistently identified functional groups of neurons (Clustering consistency in Table 6). This highlights ultra-sparse readouts as readily applicable to other architectures than CNNs, where our method still recovers functionally consistent groups of neurons.

Table 6: Performance and clustering metrics for the sparse models with a ViViT core. Values shown are means (and standard deviations for correlations) over 5 models with different seeds.

| Metrics | Unconstrained | Gumbel-Softmax |
|---|---|---|
| Predictive performance | | |
| Validation correlation | $0.538 \pm 0.014$ | $0.470 \pm 0.014$ |
| Test single trial correlation | $0.519 \pm 0.018$ | $0.450 \pm 0.014$ |
| Test correlation to average | $0.715 \pm 0.024$ | $0.630 \pm 0.021$ |
| Clustering consistency | | |
| ARI | $N/A$ | 0.523 |
| Jackknifed correlation | $N/A$ | 0.718 |

Table 7: Hyperparameters for ViV1T models of retinal ganglion cells

| Hyperparameter | Unconstrained | Gumbel-Softmax |
|---|---|---|
| ViV1T Core Architecture | | |
| Spatial Transformer Depth | 4 | 4 |
| Temporal Transformer Depth | 3 | 7 |
| Embedding Dimension | 64 | 64 |
| MLP Hidden Dimension | 128 | 128 |
| Number of Attention Heads | 4 | 8 |
| Head Dimension | 32 | 32 |
| Spatial Patch Size | $10 \times 10$ | $13 \times 13$ |
| Temporal Patch Size | 15 | 10 |
| Positional Encoding Mode | None | Learned |
| Regularization | | |
| Patch Embedding Dropout | 0.0104 | 0.0190 |
| Multi-Head Attention Dropout | 0.3812 | 0.3043 |
| Feed-Forward Dropout | 0.4159 | 0.1635 |
| Drop Path | 0.2836 | 0.2244 |
| Readout Dropout | 0.1460 | 0.0805 |
| Training | | |
| Learning Rate (Readout) | 0.0013 | 0.0028 |
| Learning Rate (Core) | 0.0033 | 0.0024 |
| Weight Decay (Readout) | 0.0001 | 0.0140 |
| Weight Decay (Core) | 0.0001 | 0.0690 |
| Adam $\beta_1$ | 0.9 | 0.8 |
| Max epochs for $\tau$ scheduler | N/A | 125 |

