# OpenReview forum: "Pick Your Channel: Ultra-Sparse Readouts for Recovering Functional Cell Types"
_ICLR.cc/2026/Conference — Submitted to ICLR 2026_

### Official Review · Reviewer_kXuj · 2025-10-30

**Soundness:** 3
**Presentation:** 3
**Contribution:** 2
**Rating:** 4
**Confidence:** 2

**Summary:**

This paper introduces ultra-sparse readouts for neural encoding models to uncover functional cell types in neural populations by enforcing sparsity in the mapping from shared visual representations to individual neuron responses. The ultra-sparse readouts combines 3 strategies: Gumbel-Softmax readout, 3D Grid readout, and REINFORCE readout. Using mouse retinal ganglion cell (RGC) and primary visual cortex (V1) datasets, the authors show that the Gumbel-Softmax readout has almost the same predictive performance as unconstrained models while naturally grouping neurons into consistent functional types. In the retina, the model identifies canonical cell classes with high internal consistency. However, performance drops in V1, which suggests that cortical neurons have a more continuous functional organization rather than discrete clustering.

**Strengths:**

1. This paper presents a method to improve the interpretability of vision-based neural encoding models. The approach is mathematically grounded and shows good empirical performance based on the reported results.

2. The analyses are detailed and comprehensive, and support the authors’ claims.

3. Improving interpretability in neural encoding models is a meaningful contribution to the field.

**Weaknesses:**

1. Although the proposed method improves interpretability, there is a notable performance drop in V1. This raises concerns about its general applicability. Ultimately, we want models that not only provide interpretability but also accurately predict neural responses to visual stimuli, as predictive performance is a prerequisite for studying how visual information is represented in the retina and the brain.

2. Although the method successfully identifies functionally consistent neuron groups, it does not directly uncover the underlying receptive field computations associated with each channel. Developing methods to this end would improve the scientific value of these interpretability methods.

**Questions:**

1. The paper uses a fixed CNN core as a proof of concept, but does the proposed method remain model-agnostic and compatible with other modern architectures, such as Vision Transformers, which are likely to be used in future large-scale studies?

2. The single-trial correlation metrics in Fig. 2 are quite low. Could the authors clarify the reason for this? For example, is it due to the use of naturalistic video stimuli rather than repeated trials? Also, why was correlation chosen as the primary evaluation metric, given that it only captures linear relationships and does not account for nonlinear dependencies or variance structure in the data?

---

> ### Author Response · Authors · 2025-11-13
> **Clarification question to reviewer kXuj**
>
> Thank you for your feedback. We believe we can address your concerns, but we have one brief question to make sure we completely understand your point and can answer appropriately. You mentioned that “Although the method successfully identifies functionally consistent neuron groups, it does not directly uncover the *underlying receptive field computations* associated with each channel.” Could you please clarify what you mean by *"underlying receptive field computations"* and the kind of answer you are looking for here?

---

> > ### Comment · Reviewer_kXuj · 2025-11-13
> >
> > I’m interested in whether this method can provide interpretability analyses beyond identifying neuron groups. For example, since the inputs are visual, is it possible to understand what visual features each functionally consistent neuron group responds to? I’m not very familiar with this field, but I’d be happy to learn more.

---

> ### Author Response · Authors · 2025-11-20
> **Authors response to Reviewer kXuj**
>
> Thank you for your review, and a quick response to our clarifying question. We’re happy to see that you found our approach “mathematically grounded” and with “good empirical performance” and “detailed and comprehensive” analyses that result in a “meaningful contribution to the field“. We think we can address your concerns about the neural computations of the channel responses, V1 performance, and clarify architectural and metric choices for our models.
>
> **Re: Although the proposed method improves interpretability, there is a notable performance drop in V1. This raises concerns about its general applicability. Ultimately, we want models that not only provide interpretability but also accurately predict neural responses to visual stimuli, as predictive performance is a prerequisite for studying how visual information is represented in the retina and the brain.**
>
> Thank you for this question. We should have included a reference that indicates that we do not expect functional cell types in V1. We have changed that in the manuscript and emphasized on this in our general response (Please also see our general response above).
> Briefly: Our method by design identifies distinct functional cell types. For V1, latest work indicates that there are no such cell types, so we expect the performance to drop.
>
> In general, we of course want these neural predictive models to perform well, and we can continue using unconstrained models for cases requiring high predictive performance. Our ultra-sparse models, instead, add interpretability, and the performance comparison between unconstrained and ultra-sparse models can be used as a diagnostic tool for the existence of functional cell types.
>
> **Re: Neural computations associated with each channel.**
>
> In Figure 5 we establish that the readout channels of the ultra-sparse model correspond to functional cell types. Each functional cell type encodes or represents a distinct visual feature.
>
> **Re: The paper uses a fixed CNN core as a proof of concept, but does the proposed method remain model-agnostic and compatible with other modern architectures, such as Vision Transformers, which are likely to be used in future large-scale studies?**
>
> Our ultra-sparse readouts theoretically remain model-agnostic due to the modular core-readout architecture of neural encoding models. This modularity allows for a free choice between CNN and Vision Transformer for the core. Indeed, we see both architectures used in similar core-readout models (Turishcheva et al. 2024, Li et al. 2023). However, you are correct that we did not show results for transformers, because we wanted to focus on the comparison to existing regularization strategies and demonstrate that the ultra-sparse readout does indeed find functional cell types.
>
> **Re: "The single-trial correlation metrics in Fig. 2 are quite low. Could the authors clarify the reason for this? For example, is it due to the use of naturalistic video stimuli rather than repeated trials? Also, why was correlation chosen as the primary evaluation metric, given that it only captures linear relationships and does not account for nonlinear dependencies or variance structure in the data?"**
>
> Correlation as a metric of predictive performance is the standard for dynamic neural encoding models (Turishcheva et al. 2024). You are right that correlation is a linear measure, but it is important to note that it is a measure between nonlinear responses and nonlinear predictions.
>
> *Correlation to Average* tests how well the model can predict a mean response (averaged across trials), thereby capturing the model’s ability to predict stimulus-specific components of the neural responses. *Single Trial Correlation* tests how well the model can predict neural responses in separate trials, which can be noisy. Single Trial Correlation includes trial-to-trial variability and cannot reach one. Hence, it is fair to expect that the model will do a better job in predicting an average response than noisy single trial responses.
>
> Finally, in the context of neuronal encoding models, our correlations values are actually quite high. For instance, state of the art for V1 prediction (e.g. Sensorium 2023 competition) achieves a single trial correlation around 0.3.
>
> **References (more in the general response above)**
>
> Turishcheva, P., Fahey, P., Vystrčilová, M., Hansel, L., Froebe, R., Ponder, K., ... & Ecker, A. (2024). Retrospective for the Dynamic Sensorium Competition for predicting large-scale mouse primary visual cortex activity from videos. Advances in Neural Information Processing Systems, 37, 118907-118929.
>
> Li, B. M., Cornacchia, I. M., Rochefort, N. L., & Onken, A. (2023). V1t: large-scale mouse v1 response prediction using a vision transformer. arXiv preprint arXiv:2302.03023.

---

> > ### Comment · Reviewer_kXuj · 2025-11-21
> >
> > Thank you for the response. I understand that the goal is to explore different approaches for learning cell types. However, from a practical standpoint, is there any empirical evidence that the proposed method generalizes to other architectures as well? For instance, another architecture might yield better predictive performance and, consequently, better interpretability, as it may explain more variance in the neural responses driven by the visual stimuli. If the authors can provide empirical results for at least one additional popular architecture, I would be happy to raise my score to 6.

---

> > > ### Author Response · Authors · 2025-11-27
> > > **Authors response to Reviewer kXuj**
> > >
> > > Upon your request, we performed additional experiments, where we empirically show that our ultra-sparse readout also works with  a transformer architecture. We have updated the manuscript with the latest results included in the last section of the Appendix, and attaching the table with main results here for convenience.
> > > | **Metrics**                        | **Unconstrained**     | **Gumbel-Softmax**    |
> > > |------------------------------------|------------------------|------------------------|
> > > | **Predictive performance**         |                        |                        |
> > > | Validation correlation             | 0.538 ± 0.014          | 0.470 ± 0.014          |
> > > | Test single trial correlation      | 0.519 ± 0.018          | 0.450 ± 0.014          |
> > > | Test correlation to average        | 0.715 ± 0.024          | 0.630 ± 0.021          |
> > > | **Clustering consistency**         |                        |                        |
> > > | ARI                                | N/A                    | 0.523                  |
> > > | Jackknifed correlation             | N/A                    | 0.718                  |
> > >
> > > In short, instead of a CNN architecture we adapted the video vision transformer core from the Sensorium Competition 2023 (Turishcheva et al. 2024) that builds on Li et al. (2023). After adapting and tuning these models with unconstrained and sparse readouts, we find slightly lower but close predictive performance to our previous CNN models (=> CNNs might be the better architecture for retinal responses given the amount of data). Similar to our CNN models, the unconstrained model outperforms the sparse model by a small margin. Crucially, however, the ultra-sparse model with the transformer core consistently identified functional groups of neurons. This demonstrates that our ultra-sparse readout generalizes across architectures.
> > >
> > > References:
> > >
> > > Turishcheva, P., Fahey, P. G., Vystrčilová, M., Hansel, L., Froebe, R., Ponder, K., ... & Ecker, A. S. (2024). The dynamic sensorium competition for predicting large-scale mouse visual cortex activity from videos. ArXiv, arXiv-2305.
> > >
> > > Li, B. M., Cornacchia, I. M., Rochefort, N. L., & Onken, A. (2023). V1t: large-scale mouse v1 response prediction using a vision transformer. arXiv preprint arXiv:2302.03023.

---

### Official Review · Reviewer_6Lyq · 2025-10-30

**Soundness:** 2
**Presentation:** 3
**Contribution:** 2
**Rating:** 2
**Confidence:** 2

**Summary:**

The paper presents a novel approach to learning ultra-sparse readouts in deep neural network models for predicting neuronal responses to arbitrary visual stimuli. This sparsity constraint minimally degrades predictive performance relative to unconstrained models, while inherently revealing functional cell types consistent with known retinal ganglion cell categories in mice. When applied to V1 neurons, the model performs worse, supporting the hypothesis of a more continuous and less discrete functional organization in V1.

**Strengths:**

The paper is scientifically motivated and proposes a conceptually novel method that bridges interpretability and performance in neural response modeling.

**Weaknesses:**

A key missing element is a clear justification of why identifying functional cell types from readout channels is important. The paper would benefit from clarifying whether the discovered functional cell types differ meaningfully from those obtained by clustering readout vectors in unconstrained models. It remains ambiguous whether the primary contribution lies in biological interpretability, computational efficiency, or predictive insight. A more explicit articulation of this contribution—and comparisons to simpler clustering-based baselines—would strengthen the paper.

**Questions:**

- In Figure 5, additional details are needed: What do the solid lines and shaded regions represent? Are these population averages and confidence intervals?

- Are the responses of known functional cell types illustrated in these plots?

- Did the method identify any novel or previously unreported cell types?

- As noted above, a more direct comparison between the sparse readout–based classification and clustering results from unconstrained models would clarify the added value of sparsity.

---

> ### Author Response · Authors · 2025-11-20
> **Author response to Reviewer 6Lyq**
>
> Thank you for your review and the constructive feedback. We’re happy to see that you found our approach “conceptually novel”, and that it “bridges interpretability and performance”. We think we can address your concerns about our intention for ultra-sparse models, the cell types it recovers and clustering baselines on readout weights.
>
> **Re: justification for sparse readout channels**
>
> We have described our motivation and contribution in the general response above (please see general response). In short: Our intention is to use sparse models with channels matched to functional cell types that we can later use to predict higher areas and understand how the complexity of neuronal processing is built up along the hierarchy.
>
> **Re: regarding Figure 5 and (un)known functional cell types**
>
> We apologize for missing the additional details in the figure 5 legend, we have updated the manuscript with this information. Solid lines represent means of all neurons grouped into the readout channel. And shaded regions are ± standard deviations.
>
> The traces in Figure 5 represent known retinal (RGC) functional cell types from Baden et al. (2016). So these are readout channel responses highly correlated with known RGC type responses. We did, however, find readout channel responses that have very low correlation (~0.1) to existing RGC cell types. They potentially represent previously undescribed responses.
>
> **Re: Comparison between the sparse readout–based classification and clustering results from unconstrained models**
>
> Thanks for this suggestion. To answer your question, we have performed Gaussian Mixture Model (GMM) and k-means clustering with the readout weights of unconstrained models. Using the cluster labels from this, we have calculated ARI scores and intracluster correlations similar to Figure 4. We report that clustering the readout weights of the unconstrained models resulted in both higher ARI (0.60 vs 0.45) and higher correlation within clusters (0.75 vs 0.65). We have updated the manuscript with the new results included in the last section of the Appendix. However, we want to emphasize that these results are still a posthoc analysis of the model and the cell types are not a part of the model.
>
> However, the above results inspired us to design yet another method to group neurons into functional cell types within the neural encoding models. Shortly, we replace the neuron-specific readout weights of the unconstrained models with mean cluster readout weights. We detail our approach in the last section of the Appendix. With that we were able to achieve the same performance as the ultra-sparse models. While the ultra-sparse model recovers functional cell types innately through end-to-end training, this additional approach still requires posthoc clustering and manual change of model parameters.

---

> > ### Comment · Reviewer_6Lyq · 2025-11-26
> >
> > Thank you for the authors’ response. It appears that the method is intended to explore alternative end-to-end approaches for identifying functional cell types, which is scientifically interesting. However, it does not seem to provide additional insight beyond what has already been shown in the literature. I will raise my score to a 4.

---

> > > ### Author Response · Authors · 2025-11-26
> > >
> > > Dear Reviewer, thank you for your response and raising the score.
> > >
> > > We would just like to clarify one point: Our intention is not a new clustering method. Clustering is a by-product that shows that our approach works.
> > >
> > > Our intention is to build a model that actually has functional cells types built into its structure. By that it becomes more interpretable. If we now plug another layer(s) on top to predict a later brain area, we can check from which channel neurons from later areas draw their information, which directly makes prediction about their functional input. Here,
> > >
> > > * we test different ways how to train such a model end-to-end,
> > > * show that the model suffers almost no loss in performance if the right training method is chosen (Gumble Softmax), and
> > > * demonstrate that the widely used approach of L1 regularization fails to identify such interpretable channels, even when they are there.
> > >
> > > We see scientific and modeling contribution in that. We would, of course, be happy if we can convince you of that, too.

---

> > > > ### Comment · Reviewer_6Lyq · 2025-11-26
> > > >
> > > > Thank you for the clarification. I think adding experiments that predict activity in a downstream brain area would be very interesting and would substantially strengthen the work. If you can demonstrate that, I’d be happy to raise the score.

---

> > > > > ### Author Response · Authors · 2025-11-28
> > > > > **Authors response to reviewer 6Lyq**
> > > > >
> > > > > We (of course) agree that applying our method to predict activity in a downstream brain area would be interesting. However, to do this with due thoroughness would be another paper and we thus believe it falls beyond the scope of the current work. The goals of which were to establish a novel method (ultra-sparse readout) that maps functional cell types natively into neural encoding models, to assess which method works best and ensure that we get reasonable results with that. We think that this by itself is a contribution that will interest others who model neuronal data via encoding models.

---

### Official Review · Reviewer_Hfcf · 2025-11-01

**Soundness:** 3
**Presentation:** 3
**Contribution:** 2
**Rating:** 4
**Confidence:** 4

**Summary:**

This paper investigates how adding sparse constraints to neural networks trained on neural data can improve the interpretability and selectivity of modeled neural responses. Specifically, it explores whether such constraints allow the network to maintain comparable performance to unconstrained models while aligning more closely with biological upstream channels. The authors analyze model performance across different visual processing stages (retina and V1), attempting to explain the differences in prediction accuracy through response continuity and channel specialization. The overall goal is to assess whether the model can learn neuron-like selectivity properties found in biological visual systems.

**Strengths:**

* The idea of incorporating sparse regularization to simulate selective neural representations provides a biologically inspired direction for neural modeling.

* The model retains comparable predictive performance despite the addition of sparse constraints, suggesting robustness and flexibility.

* Effort to connect neural network representations with retinal and cortical processing – The attempt to interpret model behavior in relation to retinal and V1 responses adds neuroscientific relevance.

**Weaknesses:**

* While the sparse constraint helps maintain similar performance, the work doesn’t convincingly demonstrate why this matters or what new insights it provides beyond showing robustness.

* The link between learned representations and specific neuronal functions (e.g., orientation selectivity, spatial frequency tuning) is vague, reducing the biological interpretability of results.

* The drop in predictive accuracy for V1 neurons indicates that the model may fail to capture hierarchical processing or contextual integration occurring in cortex.

* The authors attribute V1–retina performance differences to “response continuity,” but this reasoning feels weak and insufficiently validated.

* Both retina and V1 were modeled using the same CNN architecture, but the paper doesn’t discuss whether this choice limits representational specialization.

* The experiments do not comprehensively test how different constraints or model components affect performance and feature representation.

**Questions:**

I have serveral questions listed as belows:

Does the sparse constraint truly promote the emergence of biologically meaningful features (e.g., orientation, direction selectivity)? Can these be visualized or quantified?

Why was the same CNN architecture used for both retina and V1? Could architectural differences (e.g., receptive field size, nonlinearity) better reflect biological distinctions?

How sensitive are the results to the degree or form of sparsity applied?

Could the authors provide more evidence supporting their “response continuity” explanation for the retina–V1 performance gap?

Would more targeted ablations or neuron-type-specific analyses reveal whether the model captures functionally distinct neural subpopulations?

---

> ### Author Response · Authors · 2025-11-20
> **Authors response to Reviewer Hfcf**
>
> Thank you for your review and the constructive feedback. We’re happy to see that you found that our “robust and flexible” approach “provides a biologically inspired direction for neural modeling”, and that it “adds neuroscientific relevance”. We think we can address your concerns about biological interpretability and architectural choices for the ultra-sparse models.
>
> **Re: Biological interpretability of results. Does the sparse constraint truly promote the emergence of biologically meaningful features (e.g., orientation, direction selectivity)? Can these be visualized or quantified? Would more targeted ablations or neuron-type-specific analyses reveal whether the model captures functionally distinct neural subpopulations?**
>
> We are adding biological interpretability to the models by enforcing that neurons in a channel have to use the same features thus enforcing functional cell types. In figure 5, we demonstrate that our sparse readout approach successfully recovers known and biologically meaningful functional cell types of retinal ganglion cells described in  Baden et al. (2016). This is the reason why we included that analysis, because it directly demonstrates that the model captures known distinct neuronal subpopulations.
>
> **Re: Why was the same CNN architecture used for both retina and V1? Could architectural differences (e.g., receptive field size, nonlinearity) better reflect biological distinctions?**
>
> The CNN architecture we used in our models has previously been used for both retina (Höfling 2024) and V1 (Turishcheva et al. 2024), where it showed good performance. The exact architecture, however, is not identical for retina and V1 models, as we performed hyperparameter tuning (including nonlinearities, kernel sizes, number of layers etc. see more in Appendix) to find and later use the best performing models. We are unsure what you mean by “better reflect biological distinctions”. We demonstrate with these architectures that the model captures biological distinctions in the retina. For cortex, we believe that the drop in performance is a property of the data and no clusters should be found by the model (see answer to your last question).  Maybe you could comment on how you think they should be better.
>
> **Re: How sensitive are the results to the degree or form of sparsity applied?**
>
> We are unsure what results you are referring to. In our ultra-sparse approach we are considering the extreme case of sparsity, where each neuron is allowed to pick only one channel. In this case, we report minimal performance drop for the retinal model and successful identification of known functional cell types. Furthermore, we attempted to induce different levels of sparsity using classical L1 regularized unconstrained models (Figure 3). We found that high regularization, in an attempt to induce sparsity, leads to undesirable effects for unconstrained models, such as response shrinkage and model performance drop.
>
> **Re: Could the authors provide more evidence supporting their “response continuity” explanation for the retina–V1 performance gap?**
>
> Thanks for pointing this out. We should have provided a reference for that. As we note in our general response, our sparse readout recovers known functional cell types in retina (as it should), and does not find cell types in V1 (as it should). Since our method by design identifies distinct functional cell types, we hypothesized that V1 has continuous, rather than distinct, functional cell types. This is in line with other studies supporting the same conclusion using a clustering method based on readout weights (Nellen et al. 2025, Ustyuzhaninov et al. 2019, 2022) and even neuronal morphologies (Weis et al. 2025).
>
> **References (more in the general authors response above):**
>
> Turishcheva, P., Fahey, P. G., Vystrčilová, M., Hansel, L., Froebe, R., Ponder, K., ... & Ecker, A. S. (2024). The dynamic sensorium competition for predicting large-scale mouse visual cortex activity from videos. ArXiv, arXiv-2305.
>
> Weis, M. A., Papadopoulos, S., Hansel, L., Lüddecke, T., Celii, B., Fahey, P. G., ... & Ecker, A. S. (2025). An unsupervised map of excitatory neuron dendritic morphology in the mouse visual cortex. Nature communications, 16(1), 3361.

---

> > ### Comment · Reviewer_Hfcf · 2025-11-25
> >
> > Thank you for the rebuttal and the additional explanations. While I appreciate the clarifications, several of my original concerns remain insufficiently addressed from a machine-learning perspective.
> >
> > First, many of my questions aimed to understand the modeling implications of the sparse readout (e.g., its effect on representation structure, inductive bias, or optimization). The rebuttal mainly provides neuroscience-driven interpretations (e.g., RGC type recovery), which do not clarify the ML insight or broader modeling significance of the approach.
> >
> > Second, the explanation of the retina–V1 performance gap relies largely on biological arguments about “functional continuity”, without analyzing whether architectural choices or model-level factors might also contribute. Likewise, the justification for using similar CNN architectures across retina and V1 remains limited.
> >
> > Third, the evidence in Fig. 5 still appears qualitative: it is unclear how cell types were matched, whether ground-truth responses were quantitatively compared, or how robust the mapping is across seeds.
> >
> > Finally, the ablations remain narrow in scope. My earlier request for more comprehensive analyses (e.g., varying sparsity levels, architectural variants, or neuron-type-specific evaluations) was not fully addressed.
> >
> > In summary, although the rebuttal offers helpful clarifications, it does not resolve my main ML-oriented questions about modeling insight, architecture, ablations, and interpretability.

---

> > > ### Author Response · Authors · 2025-11-28
> > > **Authors response to reviewer Hfcf**
> > >
> > > Thank you for clarifying that your concerns are from a machine learning perspective.
> > >
> > > We would like to emphasize that we did not propose a broad applicability of the model for machine learning in general. Instead, our method and models are for neuronal data, and we see our work situated in neural system identification and interpretability. In the context of neural system identification, we do explore a soft (L1 regularization) vs. hard (ultra-sparse readout) inductive bias for neural encoding models. We strongly believe that comparing different methods and demonstrating which one works best will interest other researchers within that field.
> > >
> > > Regarding the role of architectural choices and model-level factors in explaining retina-V1 performance, we rely on prior work that: (a) establishes evidence for functional continuity (Ustyuzhaninov et al. 2019, 2022, Nellen et al. 2025) (b) establishes CNN architectures such as the one used here as standard for modeling both retina (Hoffling et al. 2024) and V1 (Turishcheva et al. 2023, Wang et al. 2025). These two lines of work already address your concerns.
> > >
> > > Regarding Figure 5, we apologize if this was not clear. We do quantitatively match the ground-truth functional cell types from Baden et al., (2016) to our readout channel ground-truth responses using maximum Spearman correlation (see Methods). We also quantify the consistency/robustness of our method in grouping neurons into functional types across model seeds in Figure 4.
> > >
> > > Regarding ablations, we do provide analyses on varying sparsity levels (Figure 3), and provide neuron-type specific analysis (Figure 5). We have now additionally shown that our method remains applicable to a video vision transformer architecture in response to Reviewer kXuj (please also see the last section of the Appendix). Hence, we do believe that we did address your main concerns. If there is any need for further clarification, we would be happy to address it.

---

### Author Response · Authors · 2025-11-20
**General response from authors**

Dear Reviewers,

Thank you for your feedback. We were happy to see that you found our work “conceptually novel”, “mathematically grounded”, with analyses that are “detailed and comprehensive”, and overall as a “meaningful contribution to the field”. Unfortunately, you also had several issues , many of them related  to the fact that we did not explain the context and motivation of our work properly to a general enough audience. We would like to make up for that and explain it here. We will post responses to your specific points in the responses to each of you separately.

We would appreciate it if you could indicate whether this explanation would make you reconsider the low rating for this paper, given that we appropriately update the manuscript and address your other issues as well.

**Motivation**: Predictive models of neuronal activity are often clustered into “task-driven” and “data-driven”. In task-driven models, representations from deep networks pre-trained on a particular task are used to predict neuronal activity. There, the focus is on the representations learned from the task and prediction of neuronal activity is used to “align” the representation to the brain area. In contrast, data-driven models (which we use here) are directly trained from scratch to predict neuronal activity and are used as “digital twins” to learn image- and video-computable response functions of neurons from data. These models have been used extensively in prior work to learn about the receptive field and coding properties of neurons (see e.g. Walker et al. 2019, Franke et al. 2024, Höfling et al. 2024).

In some works, the readout vectors of these models are used as functional fingerprints because they  contain the condensed information about the functional properties of the neurons (see e.g. Wang et al. 2025 , Ustyuzhaninov et al. 2019, 2022).

One question of interest in biology is the existence of functional cell types. Functional cell types can be defined as groups of neurons that are functionally similar in how they process visual stimuli. Or, in other words, different functional cell types encode distinct visual features (Baden et al. 2016).

In the retina the functional cell types are usually found by clustering the responses to an informative stimulus, such as temporally changing full field flicker (chirp) and a moving bar. In V1 readout vectors of digital-twins have been used to search for functional cell types (Ustyuzhaninov et al. 2019, 2022). However, evidence currently suggests that there are no distinct cell types in V1 and neurons rather form a continuum of functions (Nellen et al. 2025).

Current clustering methods mostly perform a posthoc analysis, extracting readout weights from an already trained model and analyzing them (by clustering, embedding or classification). Nellen et al. (2025) recently combined the clustering of readout weights with model training. In all cases, however, the clustering is done on the already learned representations (readout weights) of the model. Importantly, in all previous models, neurons still have their “own” readout vector and are not forced to share the same vector with other neurons of the same cluster. In that sense, the functional cell types are **not** baked into the model. This is what we want to address with our work.

We want to generate a model that has components (channels) that are directly identifiable with cell types. Thus our main goal is to get a model that has interpretable elements. Our intention is to use such a model later to predict higher areas and understand how the complexity of neuronal processing is built up along the hierarchy.

What we explore in the paper is different ideas on how to get a model to learn such cell types (L1 regularization, extension of the Gaussian readout, or picking channels by REINFORCE or Gumbel Softmax). We demonstrate that the standard method of L1 regularization, used in almost all previous works, fails at that. We also demonstrate that the Gumbel Softmax solution, we propose, fixes this issues, and correctly finds cell types in the retina (as it should) and correctly does **not** find cell types in V1 (as it should).
Thus our contribution is to show how to build a model with interpretable components which could not be achieved by current ways (regularization).

We hope this answers your questions about the motivation behind this work. If it is still unclear, we are happy to elaborate further.

---

> ### Author Response · Authors · 2025-11-20
> **References for the general response from authors**
>
> References:
>
> Walker, E. Y., Sinz, F. H., Cobos, E., Muhammad, T., Froudarakis, E., Fahey, P. G., ... & Tolias, A. S. (2019). Inception loops discover what excites neurons most using deep predictive models. Nature neuroscience, 22(12), 2060-2065.
>
> Franke, K., Cai, C., Ponder, K., Fu, J., Sokoloski, S., Berens, P., & Tolias, A. S. (2024). Asymmetric distribution of color-opponent response types across mouse visual cortex supports superior color vision in the sky. elife, 12, RP89996.
>
> Höfling, L., Szatko, K. P., Behrens, C., Deng, Y., Qiu, Y., Klindt, D. A., ... & Euler, T. (2024). A chromatic feature detector in the retina signals visual context changes. Elife, 13, e86860.
>
> Wang, E. Y., Fahey, P. G., Ding, Z., Papadopoulos, S., Ponder, K., Weis, M. A., ... & Tolias, A. S. (2025). Foundation model of neural activity predicts response to new stimulus types. Nature, 640(8058), 470-477.
>
> Ustyuzhaninov, I., Cadena, S. A., Froudarakis, E., Fahey, P. G., Walker, E. Y., Cobos, E., ... & Ecker, A. S. (2019). Rotation-invariant clustering of neuronal responses in primary visual cortex. In International Conference on Learning Representations.
>
> Ustyuzhaninov, I., Burg, M. F., Cadena, S. A., Fu, J., Muhammad, T., Ponder, K., ... & Ecker, A. S. (2022). Digital twin reveals combinatorial code of non-linear computations in the mouse primary visual cortex. BioRxiv, 2022-02.
>
> Baden, T., Berens, P., Franke, K., Román Rosón, M., Bethge, M., & Euler, T. (2016). The functional diversity of retinal ganglion cells in the mouse. Nature, 529(7586), 345-350.
>
> Nellen, N. S., Turishcheva, P., Vystrčilová, M., Sridhar, S., Gollisch, T., Tolias, A. S., & Ecker, A. S. (2025). Learning to cluster neuronal function. arXiv preprint arXiv:2506.03293.

---

### Comment · Area_Chair_KA1m · 2025-11-25

Dear Reviewers,

This is a gentle reminder to please take a moment to review the authors’ rebuttal for the manuscript currently under your evaluation. Your timely feedback will help us proceed with the next steps in the review process.

Thank you for your time and assistance.

Best regards,
AC

---

### Author Response · Authors · 2025-12-03
**Summary of author-reviewer discussion**

We thank the reviewers for engaging in a productive discussion. Initial reviews found our work “conceptually novel”, “mathematically grounded”, with analyses that are “detailed and comprehensive”, and overall as a “meaningful contribution to the field” that “provides a biologically inspired direction for neural modeling”. The reviewers raised issues around the following questions:
* clarification of context and motivation of our work (**6Lyq**)
* performance drop in V1 compared to retina (**kXuj**, **Hfcf**)
* architectural choices (**kXuj**, **Hfcf**) relating to model-agnosticity of our ultra-sparse readout method
* biological insight from ultra-sparse readouts (**6Lyq**, **kXuj**)

In response to the issues, we provided detailed clarifications about the motivation and biological insight of ultra-sparse readouts in our general response, in particular providing references to previous work establishing that V1 performance drop is expected as a consequence of the organization of mouse V1 and not as a shortcoming of our method.

In addition, we have provided additional experimental results in response to reviewer **kXuj** where we successfully applied our ultra-sparse readouts to a video vision transformer architecture. This empirically showed the model-agnostic applicability of our modular method.

We have also conducted additional experiments in response to reviewer **6Lyq** showing direct comparison of clusters recovered using ultra-sparse readout vs unconstrained readout weights. This discussion inspired us to design yet another method to group neurons into functional cell types within the neural encoding models. We have updated the manuscript during the discussion phase with these additional experimental results.

We believe the discussion strengthened our work without changing its main contribution, and addressed the issues raised by the reviewers. However, not all reviewers agreed to that. When the discussion was interrupted, the discussion ended in the following state:

* Reviewer **Hfcf** questioned the ML significance of sparse constraints, wanted clearer biological interpretability, criticized "response continuity" explanation for V1 performance gap, and requested more comprehensive ablations. After the rebuttal, **Hfcf** remained skeptical from an ML perspective. We argue that (i) our work is intended for neural system identification (not general ML; we never claimed that), (ii) we did provide reference to previous work for the response continuity argument, and that (iii) we already provide interpretable and quantitative results in our paper. In addition, the ViT experiments for reviewer **kXuj**, added evidence for a broader applicability.
* Reviewer **6Lyq** criticized the missing justification for why sparse readout channels matter and how they differ from post-hoc clustering approaches. The reviewer raised the score to 4 after we added GMM/k-means comparisons and clarified our motivations. During the discussion, the reviewer suggested the work needs downstream brain area prediction experiments to substantially strengthen its justification – possibly increasing the score further. We argued that this would go beyond the scope of the current paper as it would be an additional study requiring additional results in already limited space.
* Reviewer **kXuj** was concerned about the V1 performance drop, and the generalizability to modern architectures. We addressed concerns about V1 prediction performance by providing references to work showing that distinct functional clusters are not to be expected in mouse V1, justified the correlation metrics, and added ViT experiments to address the reviewer’s concerns. Our results show generalization across architectures. The reviewer initially stated willingness to raise the score to 6 based on these additional (ViT) experiments. However, the raise never happened, possibly due to the incident around the OpenReview Bug.

In summary, we sincerely appreciate the time and effort all reviewers have invested in evaluating our work. While we respect their perspectives, the discussion ended in disagreement about the contribution of our work. Our primary goal is to advance neural system identification by developing encoding models where functional cell types are intrinsic to the architecture rather than discovered post-hoc. We demonstrate that widely-used approaches (L1 regularization) fail at this task, propose an alternative that succeeds (Gumbel-Softmax readout), and quantitatively validate it against known cell types in the retina. We still believe that this represents a meaningful methodological contribution to computational neuroscience, particularly for researchers seeking interpretable models that respect biological organization even if it may not address broader machine learning applications that some reviewers were seeking.

---

### Meta-Review · Area_Chair_crY5 · 2026-01-07

**Summary:**

The paper proposes to learn ultra-sparse readout for neural response prediction models. While the approach is technically interesting, there are several concerns from reviewers. The contribution is not clearly articulated, as it remains unclear why single-channel readouts provide new insights beyond simpler baselines such as clustering readout weights from unconstrained models. The biological interpretability of the learned channels is weak, with only limited and qualitative links to known neuronal properties. Reviewers also point out that the substantial performance drop for V1 neurons raises concerns about generality and suggests the method may fail to capture cortical computations. The work prioritizes interpretability over predictive accuracy without demonstrating clear biological or computational benefits.

**Reviewer Concerns:**

While the authors made a substantial effort during the discussion to clarify their motivation, add experiments, and better position their contribution, the core concerns raised by multiple reviewers remain only partially resolved. In particular, key questions about the necessity of ultra-sparse readouts beyond clustering or existing interpretability methods were not conclusively answered, and the biological insights enabled by the approach remain limited and somewhat indirect.

**Reviewer Scores:**

As reflected in the discussion, although one reviewer raised their score to the acceptance threshold, at least one reviewer remained skeptical from a machine learning perspective. Despite constructive clarifications and incremental improvements, the discussion did not reach a consensus that the work delivers sufficiently strong biological or methodological advances to meet the acceptance bar.

---

### Decision · Program_Chairs · 2026-01-26

Reject